**Chicxulub Museum, Geosciences in Mexico, Outreach and Science Communication - Built From the Crater Up**

Jaime Urrutia-Fucugauchi[1,2], Ligia Perez-Cruz[1,2,3] Araxi O. Urrutia[4,5]

[1] Programa Universitario de Perforaciones en Océanos y Continentes, Instituto de Geofísica, Universidad Nacional Autónoma de México, Coyoacan 04510 Mexico, Mexico
[2] Instituto de Investigación Científica y Estudios Avanzados Chicxulub, Parque Científico y Tecnológico de Yucatán, Sierra Papacal, Merida 97302, Yucatán, Mexico
[3] Coordinación de Plataformas Oceanográficas, Coordinación de la Investigación Científica, Universidad Nacional Autónoma de México, Coyoacan 04510 Mexico, Mexico
[4] Milner Centre for Evolution, Department of Biology and Biochemistry, University of Bath, Bath BA2 7AY United Kingdom
[5] Instituto de Ecología, Universidad Nacional Autónoma de México, Coyoacan 04510 Mexico, Mexico
Correspondence: juf@geofisica.unam.mx

## Abstract

The Chicxulub science museum is special, in that it is built around an event in geological time representing a turning point in the planet`s history and which brings together the Earth system components. The studies on the Chicxulub impact, mass extinction and Cretaceous/Paleogene boundary provide an engaging context for effective geoscience communication, outreach and education. The museum is part of a research complex in the Yucatan Science and Technology Park in Mexico. Natural history museums with research components allow the integration of up-to-date advances, expanding their usefulness and capabilities. The impact ranks among the major single events shaping Earth's history, triggering global climatic change and wiping out ~76% of species. The ~200 km Chicxulub crater is the best preserved of three large terrestrial multiring impact structures, being a natural laboratory for investigating impact dynamics, crater formation and planetary evolution. The initiative builds on the interest that this geological site has for visitors, scholars and students by developing wide-reaching projects, a collaboration network and academic activities. The Chicxulub complex serves as a hub for multi- and interdisciplinary projects on the Earth and planetary sciences, climate change and life evolution, fulfilling a recognized task for communication of geosciences. After decades of studies, Chicxulub impact remains under intense scrutiny and this program with the core facilities built inside the crater will be a major player.

Keywords: Chicxulub science museum, Chicxulub impact, End-Cretaceous mass extinction, Geosciences communication, Mexico

## 1. **Introduction**

Geosciences in Mexico has a rich tradition that can be traced to the Mesoamerican cultures. Considering the intense tectonic, seismic and volcanic activity, energy and mineral resources and diverse geological record, the geosciences play minor roles in the social, academic and political discussion. Addressing this situation requires the development and implementation of effective geoscience communication and education programs. The Chicxulub science museum developed around a unique geological event that marks the transition of the Mesozoic and Cenozoic Eras provides the context for a major program, which is based on the studies of the End-Cretaceous mass extinction, Chicxulub impact and Cretaceous/Paleogene (K/Pg) boundary.

Natural history and geological museums have a rich tradition in which collections of rocks, minerals, meteorites and fossils, play an important role in non-formal education, with high learning potential for students, museum-school synergies, science engagement, and teachers' professional development (Stevenson, 1991; Allen, 2004; Panda and Mohanty, 2010; Dahlstrom, 2014; Mujtaba et al., 2018). Museums with research departments allow the integration of scientific advances, taking advantage of thematic exhibits, interactive displays and virtual reality experiences (Collins and Lee, 2006; Panda and Mohanty, 2010; Louw and Crowley, 2013).

Field trips to geological sites are important components of the university educational programmes and professional workshops, meetings and congresses. National parks, Global Geoparks and UNESCO heritage natural sites attract large numbers of scholars and students as well as visitors. Museums of natural history, geology and mineralogy present exhibits related to life evolution, fossil record, planetary exploration, plate tectonics and meteorite impacts (MacFadden et al., 2007; Koeberl et al., 2018). Some, such as the Smithsonian National Museum of Natural History, the British Museum, Geological Museum of China, Museum of Natural History of Paris, Natural History Museum in Vienna, and Geological Museum of Barcelona, among many others, have rich fossil, meteorite and mineralogical collections (Komorowski, 2006; Koeberl et al., 2018). Geological site and crater museums are less numerous and include the Ries crater Museum in Nôrdlingen, the Meteor Crater Museum in Arizona, the Tswaing Crater Museum in South Africa, the Steinheim Crater Museum in Germany, and the Meteorite Museum at Rochechouart (Pôsges, 2005; Buchner and Pôsges, 2011).

The Chicxulub complex (CIRAS) with the science museum, laboratories and core repository is
housed in the Yucatan Science and Technology Park (PCYTY), southern Mexico (Figs. 1 and 2).
The museum builds on the interest generated by the Chicxulub impact and the K/Pg mass
extinction, which includes the dinosaurs, ammonites, marine and flying reptiles among many
organisms. Here we outline how it has developed and examine the potential that a facility built
around an attractive unique event and geological site offer, including the challenges ahead.
Understanding Earth`s origin and evolution, geologic time, tectonic processes, rock and fossil
record, life evolution and extinction presents challenges that have been considered in designing
the Chicxulub exhibits and activities. The link to research permits interactions of researchers and
students with visitors through conferences, seminars and workshops, and visits to the laboratories
and geological sites. How this translates in better appreciation and understanding of Earth and
planetary sciences on science communication is a major part of the planning. Other issues on
popular attractive themes like meteorite impacts and dinosaur extinction provide attractive
contexts for geoscience engagement.
2. **Geosciences in Mexico**
Research groups in Mexico have developed public outreach projects on hazard risks, climate
change, mineral and energy resources, renewable energy and environmental impacts.
Nevertheless, we are yet to have long-term programs and effective influence on the education
system, policy decision-making and society.
The number of researchers is small compared to the size of the country and the economy, which
is a limitation shared with the developing countries. The Earth sciences system focuses on basic
and applied projects has expanded over the past years (Atlas Ciencia Mexicana, 2012). The
Seismological Survey, Geomagnetic Observatory and Tidal Service are operated by the National
University of Mexico. Federal institutions include the Mexican Geological Survey, National
Institute of Information and Statistics, National Center of Disaster Prevention, National
Metereological Service, National Water Commission and Tidal Service, which carry out the
cartography and the instrumental networks. The Oil Company Petroleos Mexicanos (Pemex) and
the Petroleum Research Institute conduct marine and on land oil and gas exploration and
production projects.
The geosciences program aims to develop a strategy linking research, policy makers and society,
with the Chicxulub complex providing the physical and human capacities that allow the expansion
of objectives, capacity-building, outreach, educational and operational activities. Coordinated
projects and policy-decision initiatives are needed, including those on disaster prevention and
mitigation, climate change, land management, sustainable programs, country-wide geophysical
surveys, renewable energy resources, oceanographic and marine geophysical surveys and
monitoring instrumental networks.
With the globalized economy, population growth and demographic changes, the demand on energy
and mineral resources has increased worldwide. In parallel, climate change, earthquakes, volcanic
eruptions, tsunamis and hydrometereological phenomena, contamination, deforestation,
extinctions and sea-level rise affect the societies. The transformation from free-market societies to
the knowledge societies, based on and driven by science and technology highlights the role of
geosciences internationally. In countries like Mexico where energy and mineral resources are
major components of the economy and where geophysical phenomena pose risks to the population,
geosciences might be expected to be the country`s priorities. This is not the case, which emphasizes
the need for informing decision makers and society on the role of geosciences on the planet
conservation and sustainable development.
International programs open collaboration opportunities for developing countries. Mexico has
participated in international programs like the International Geophysical Year, Polar Year, Upper
Mantle, Geodynamics, Lithosphere (ILP), International Ocean Discovery Program (IODP),
International Continental Drilling Program (ICDP) and Geosphere-Biosphere program. It recently
formed part of the United Nations International Year of Planet Earth (IYPE), International Council
of Science ICSU Future Earth program, and UNESCO geosciences programs. Our program linked
to these initiatives integrates the Chicxulub drilling and geophysical surveys and the participation
in IODP, ICDP, IYPE and ILP projects.

3.  **Chicxulub Impact and Mass Extinction**

The Chicxulub impact marks a major event shaping life on Earth (Alvarez et al., 1980; Schulte et
al., 2010). Impact marks the end of the Mesozoic Era, with the mass extinction wiping out ~76%
of species including dinosaurs, ammonites, marine and flying reptiles, and the start of the Cenozoic
that saw important radiations of many groups including mammals and birds. The Chicxulub
structure formed by an asteroid impact on the Yucatan carbonate platform in the southern Gulf of
Mexico was first identified in the Pemex exploration surveys and drilling programs (Penfield and
Camargo, 1981). Chicxulub is a complex crater with a ~200 km rim diameter (Fig. 3), which has
been investigated by an array of geophysical/geological surveys and drilling programs (Fig. 4;
Hildebrand et al., 1991, 1998; Sharpton et al., 1992; Urrutia-Fucugauchi et al., 2008).
The K/Pg boundary is marked globally by the impact ejecta layer, characterized by the iridium and
platinum group elements derived from the impacting body (Fig. 4c; Schulte et al., 2010). The
impact and its effects on Earth's climate and life evolution have been intensively studied (Alvarez
et al., 1980; Mukhopadhyay et al., 2001; Schulte et al., 2010; Urrutia-Fucugauchi et al., 2011;
Lowery et al., 2018). Impact had short- and long-term global effects on the climate and
environment, providing lessons for understanding the impact of man-made greenhouse emissions.
Although the mechanisms for the extinction and subsequent species diversification remain under
scrutiny, studies of this mass extinction uncover general principles governing species/clade
resilience and evolvability in response to rapid climate and environmental changes. So, in
summary, Chicxulub presents an opportunity to showcase the holistic and integrated nature of
Earth system science.
4.  **Background and Development of Chicxulub Museum**
The CIRAS research and museum facilities are housed over an area of ~19 square kilometers
located in the central sector of the Yucatan Science and Technology Park (Figs. 1 and 2). The
CIRAS is a joint project between the National University of Mexico, the National Council of
Science and Technology and the Ministry of Science and Higher Education of the Yucatan
government that has developed for a decade.
From the initial phases, the plan included the site museum on the Chicxulub impact and its effects
on the planet and life evolution. The first phase was completed in 2011 with the Chicxulub
Museum housed in the second and third floors of the PCYTY Central Library (Fig. 5). The second
phase was the Chicxulub exhibition in the Meteorite Hall of the Grand Museum of the Maya World
(Gran Museo de Mundo Maya) in Merida City (Fig. 6). Inaugurated on December 12, 2012, the
Chicxulub exhibition was awarded the 2013 Miguel Covarrubias Prize from the National Institute
of Anthropology and History
The Chicxulub exhibition in the Grand Museum of the Maya World attracted large numbers of
visitors, students and researchers. The Chicxulub Impact and Extinction of Dinosaurs exhibition
was planned at the time of the Mayan prophesy of the end of the world and included displays on
historical accounts of catastrophic prophesies of various cultures. The exhibition addressed beliefs
on celestial phenomena such as comets and lunar and sun eclipses, which in some societies were
associated with catastrophes, diseases, warfare and social unrest. The contrasting views were
presented in the framework of the Chicxulub impact, extinction of dinosaurs and other species and
the end of the Mesozoic Era.
Museum visits start with a video presentation on the Chicxulub impact and mass extinction,
followed by introductions to comets, asteroids and meteorites, early observations of meteorite falls
and cometary showers and how they evolved as part of the studies of the planetary system. A major
component is the exhibits of the fossil record, geologic time and evolution of the dinosaurs, marine
microorganisms, ammonites and flying and marine reptiles. Initial Chicxulub studies were linked
to oil exploration in southern Mexico and the geological characteristics of the Yucatan peninsula
(Urrutia-Fucugauchi et al., 2013). Exhibits display surface geological processes, with the aquifer,
groundwater flow and fracturing influenced by the buried crater, which can be traced by the ring
of cenotes and semicircular topographic depression over the crater rim. Related programs at the
museum are the conferences, seminars and symposia, including the progress reports of the research
and drilling projects.
The PCYTY Chicxulub Museum has attracted large numbers of visitors. Entrance is free and
records are only for the guided tours and appointed visits of school children. In four years, number
of visitors is around seventeen thousand, including six thousand school students and one thousand
pre-school children. The visitors to the Chicxulub Exhibition at the Grand Museum have been
more numerous, due to its association to the archaeological exhibits and easy access in Merida
City. Comments and response discussed below come mainly from the student groups and teachers,
with additions from groups during conferences and seminars. The guided tours for school groups
offered the advantage of engaging with the teachers, which provided valuable interactions and
feedback. In connection with the museum exhibits, conference series and workshops were held
with the participation of students and researchers. Among them, the workshops of the drilling and
marine geophysics projects and on geosciences education.
Around the initial plan, research facilities expanded to include laboratories and the core repository
built in the Yucatan Science Park, which houses academic and research institutions, start-ups and
research-oriented firms, including Yucatan State University, National University of Mexico, and
National Council of Science centres. CIRAS construction project took several years with the center
formally established in February 2018 with the inauguration of the laboratories and core repository
(Fig. 7). It has access to the National Hydrocarbon Core Repository and the apartment blocks to
host visiting academics and students. Third phase started in 2016 with construction of the larger
museum facility that started operating in the early 2019.
5. **Chicxulub Complex**
**5.1 Science Museum**
Studies on large meteorite impacts, dinosaurs, mass extinctions and life evolution attract the
interest of wide audiences, opening interesting possibilities for science communication. The
exhibits are organized around the studies of the Solar System, impact cratering, evolution of
planetary surfaces, Chicxulub impact, crater formation, impact effects on climate and life-support
systems, extinction of organisms, biotic turnover and life evolution. Exhibits aim to present,
inform, engage and entertain visitors through studies on the Chicxulub impact, life evolution, K/Pg
turnover and related inter- and multidisciplinary research (Figs. 9 and 10).
Exhibits on the Universe hall present an introduction to the origin and evolution of the Universe,
the formation of stars and galaxies, the Milky Way galaxy and the Solar System. The formation of
planetary systems involves dynamic processes with collisions at different scales, with formation
of first solids, planetesimals and large bodies. The origin and evolution of planetary systems are
marked by collisions of bodies, which are the main process in the formation of planets, satellites,
dwarf planets, asteroids and comets. Impact craters characterize the surfaces of solid planetary
bodies and constitute the geological record of the dynamic evolution through time and space.
The hall on the Solar System and Impact Cratering presents an engaging introduction on the
characteristics and evolution of planetary surfaces, impact dynamics, crater formation, impacts on
time and space, comets, near-Earth asteroids and impact hazards. Hypervelocity impacts deliver
high amounts of energy in short time scales, resulting in deep excavation cavities, material
transport and deformation. Planetary surfaces preserve records of impacts, with the magnitude and
frequency of impacts higher in the early stages. Impact cratering is a major process in the evolution
of planetary surfaces and deep interiors. The terrestrial record has been erased and modified, with
a limited number preserved in contrast to other bodies like the Moon, Mars, Venus and Mercury.
The exhibits on the Chicxulub structure introduce the crater and impact effects. The large multiring
crater is the best preserved of the three large impact structures in the terrestrial record, being a
laboratory for investigating impact dynamics, crater formation and planetary surface evolution
(Melosh, 1989; Urrutia-Fucugauchi and Perez-Cruz, 2009). The structure is located half on land
and half offshore with geometric center at Chicxulub Puerto on the coastline.
The hall on the End-Cretaceous extinction and life evolution introduces the effects of the meteorite
impact on the life-support systems, linking the impact with the mass extinction. Exhibits introduce
the fossil record, geological processes, the geological time scale and concepts of deep time and
life evolution. The mass extinction marks the boundary between geological eras, which in the
geological record is marked by the Chicxulub ejecta layer. Interactive exhibits introduce macro-
evolutionary trends, with species communities and diversification after the impact.
Exhibits include challenging themes on life evolution, extinctions, emergence of species,
macroevolution and climate change (Sepkoski, 1998; Jablonski, 2006, 2008). Experiences in
natural history and science museums emphasize the roles of teachers and museum staff in
interacting with visitors, particularly with school groups and students on difficult topics. This is
the case with exhibits on the End-Cretaceous mass extinction and asteroid impact effects that
permit issues such as present-day global warming, environmental problems and extinctions to be
addressed.
The Museum includes an auditorium, meeting rooms and a projection room, used to present videos
and animations of the Chicxulub impact; plus a children playing room. Independently managed
coffee shop and souvenir stores complement the facilities. The museum has spaces to host
temporary exhibits on the Gulf of Mexico-Caribbean Sea, mineral and energy resources, global
climate change and biodiversity, which open collaboration programs with other institutions.
Spaces around the museum incorporate outdoor exhibits (dinosaurs and marine and flying reptiles)
that take advantage of the vegetation with endemic plants and large-size fossiliferous carbonate
rock boulders (Fig. 10). Additionally, the PCYTY Botanical Garden is next to the museum, which
is open for join activities.

## 5.2 CIRAS Research Areas

The institute, core repository and six laboratories have analytical facilities for core analyses, sample preparation, petrography, micropaleontology, geochemistry and physical properties. Laboratories are equipped with core scanners, X-ray fluorescence system, gamma-ray core logging system, magnetic susceptibility meters, electrical resistivity meter, petrographic microscopes, laser particle analyser and electronic scanning microscope (e.g., Fig. 7). The core repository has facilities for conducting experiments, slim-core logging sensors and geophysical instruments, including gravity, resistivity and magnetic field meters.

Ongoing projects focus on studies of crater structure, dimensions, morphology, breccia deposits, melt sheet, target deformation, impact-induced hydrothermal system, pre-impact structures and post-impact processes. Chicxulub has been investigated with a wide array of geophysical methods, including gravity, magnetics, electromagnetics and seismic reflection (Hildebrand et al., 1998, Sharpton et al., 1993; Collins et al., 2008; Urrutia-Fucugauchi et al., 2011; Morgan et al., 2016).

The structure and ejecta are not exposed, making drilling an indispensable tool to sample the impactites and pre- and post-impact sedimentary rocks (Fig. 3). Initial drilling was carried out by Pemex oil company with intermittent core recovery providing samples that were key for confirming the age of the impact structure (Hildebrand et al., 1991; Sharpton et al., 1992). Subsequent drilling programs incorporated continuous core recovery and geophysical logging (Fig. 4; Urrutia-Fucugauchi et al., 2004, 2008), with tens of thousands of core samples distributed to groups in different countries, which has allowed to expand the research on the crater and K/Pg boundary.

Studies investigate impact effects on climate and life support systems (Alvarez et al., 1980; Schulte et al., 2010; Urrutia-Fucugauchi and Perez-Cruz, 2016; Lowery et al., 2018), with recent ones shedding light on factors determining the likelihood of taxa becoming extinct as in the case of arboreal birds after forests disappeared (Field et al. 2018). Mass extinction was followed by radiations in numerous taxa including mammals (Dos Reis et al. 2012), worm lizards (Longrich et al. 2015) and birds (Field et al., 2018). Further understanding of the factors driving species extinction and radiations is crucial to make predictions on the effects of man-induced climate changes.

CIRAS carries research relevant to the communities in Yucatan, studying the low relief karstic
terrains (Fig. 3). The city of Merida, located ~30 km away from the coastline, is just a few meters
above sea level. The platform is an extensive low-inclination shallow ramp, which records the sea-
level fluctuations during the Late Pleistocene glaciation and the Holocene. Yucatan is in the
trajectory of hurricanes and tropical storms, with a thin soil cover, no surface waters and vulnerable
to coastal erosion, marine intrusion, aquifer contamination and global warming with changes of
precipitation, sea level, cloud coverage and evaporation.
The northern Yucatan peninsula is marked with sinkholes and dissolution structures and the buried
structure exerts a strong influence in surface geological processes including subsidence, fracturing,
groundwater flow, coastal and karst processes. The density and distribution of karstic structures
relate to dissolution and in turn to fracturing, topography, rainfall and groundwater flow. The
sinkhole distribution correlates with the buried structure, notably with the cenote ring located over
the crater rim. Surface fracturing is related to the stress/strain state, with the regional tectonics,
differential subsidence of the crater fractured breccias and carbonates surrounding the crater and
rheological properties of the surface formations. Coastline morphology and processes are related
to the buried structure, marked by the correlation at the intersections with the gravity anomaly
rings. The thick carbonate cover has protected the structure and ejecta deposits from erosion, while
adding challenges for the studies. The structure, characterized on the surface by gravity and
magnetic semi-circular concentric patterns (Fig. 3), is characterized by a gravity high and high-
amplitude magnetic anomalies associated with the basement uplift, peak-ring and impactite
deposits. The crater rim and terrace zone are marked on the surface by the cenote ring, fracturing
and semi-circular topographic depression.
6. **Discussion**
In Mexico, research projects address societal issues such as hazard risks, climate change, mineral
and energy resources, renewable energy and environmental problems, but geoscientists have yet
to have long-term programs that have an effective public, educational and policy impact. The
CIRAS offers a potential facility for doing that.
It forms a collaboration network and program hub to carry research on the Chicxulub crater and
relations to life evolution, impact dynamics and cratering and the effects on planetary scales. As
such, it develops from the studies of a unique event marking a turning point in the planet`s

evolution, thus offering interesting opportunities and challenges. How is the program addressing and developing its capabilities for outreach, education and geoscience communication? How attractive is this unique geological site for engaging visitors? How are concepts such as nature of geologic time, life evolution, fossil record, climate change introduced? How do visitors respond to exhibits and related activities?

The mass extinction provides an engaging start point and context for addressing planetary evolution and how life evolves linked to geological processes, climate and environment. This permits introducing fundamental concepts on geological time, processes, life evolution, Earth System, interconnections and role of sudden changes.

**6.1 Outreach and Education**

Mujtaba et al. (2018) reviewed the learning potential of natural history museums, focusing on school students, interactions museum-schools, science engagement and teachers' professional development. They have a rich tradition, with exhibits, interactive displays and collections of rocks, minerals, fossils and animals and plants, playing important roles in the conservation and preservation of fossils, minerals and geological sites (Lipps and Granier, 2009; Boonchai et al., 2009). Natural history exhibits and interactive displays on life evolution permit addressing difficult concepts that include natural selection, speciation, extinction, concepts of deep time, intense sudden high-amplitude events versus gradual incremental changes, global versus local processes and macroevolution (Baum et al., 2005; Diamond and Scotchmoor, 2006; Spiegel et al., 2012; MacDonald and Wiley, 2012). Visitors to natural history museums are in general more familiar with evolutionary concepts than those who do not have the experience. Studies on how visitors view, approach and accept/reject/ignore evolution show that those with museum experiences are more familiar with life evolution than the general public (Mujtaba et al., 2018). However, large sectors face difficulties comprehending those concepts, which is the case with topics such as climate change, global warming and anthropogenic impacts.

In the Chicxulub museum, activities include conferences, seminars, drawing contests for school children in primary schools, material/publications, interaction with teachers and schools. Two GIFT (Geosciences Information for Teachers) Workshops of the European Geosciences Union (EGU) have been held in Merida in 2010 and 2016. The GIFT Workshops were organized in collaboration with the Secretaries of Education and SIIES, the Mexican Academy of Sciences and

scientific societies. The Panamerican GIFT Workshop of the EGU capacity-building program
scheduled for October 2020 in the Chicxulub Museum has been postponed for 2021.
The field experiences take advantage of museum location, to enhance learning experiences from
field observations of rocks, fossils and local flora and fauna. The PCYTY Botanical Garden with
marine fossil-rich outcrops further expands the museum experience. Additional activities include
microscopic observations for petrographic and microfossil analyses, complementing activities in
the classrooms and museum visit. Novel avenues use the internet, digital tools, smartphone
applications and new spaces particularly for the natural and physical sciences (e.g. Braund and
Reiss, 2004, 2006). Field trips to K/Pg boundary sites open opportunities to understand impact
effects and impact geological record (Fig. 6). The nearest sites in Campeche, Quintana Roo and
Belice are displayed in exhibits, maps, videos and images, and complemented by animations
illustrating how ejecta was emplaced proximal in the proximal area and at distant locations.
**6.2 Challenges and Approaches**
The crater and proximal ejecta deposits are not exposed at the surface, which is a challenge in
comprehending the huge size and characteristics of the structure. We also found that visitors have
difficulties understanding how and why dinosaurs went extinct, dynamics of asteroid impacts and
crater formation, sequence of events, other species affected, what happened with the mammals,
why and how some mammal species did not go extinct, how some species went extinct while
others do not. The Chicxulub size and relation of buried structure to the ring of cenotes are difficult
to appreciate because of the large dimensions. Following the sequence of events and crater
formation in a short time and with large energy release also generates questions. For instance,
many visitors consider that impact formed the cenotes (particularly the cenote ring), though they
acknowledge the crater lies deep beneath and that the cenotes are recent surface features. The
origin of Chicxulub structure also generates confusion, though there are exhibits on the craters on
the Moon and other bodies, visitors have difficulty understanding impact craters and volcanic
craters as formed by different geological processes.
Presenting in an engaging way concepts on geological time, evolution, fossil record and geological
processes is no easy task. Museums have developed different approaches, with results showing
mixed responses and the complexities of the subjects (Braund and Reiss, 2004, 2006; Allen and
Gutwill, 2004; MacFadden et al., 2007; Mujtada et al., 2018). In the museum, different approaches
are tied around attractive issues. For instance, exhibits on dinosaurs attract more interest than
displays on other groups, so they are taken to engage visitors. Widespread interest in dinosaurs
comes from their large sizes and diversity, including the giant sauropods, predators like the T. rex
and raptors and the feathered dinosaurs. Long-term evolution and adaptations are introduced by
showing how successful were the dinosaurs during the Mesozoic, occupying the ecosystems in the
continental land masses including the polar regions (Sereno, 1999; Barret et al., 2009).
Mammals are also attractive, particularly those on the Late Pleistocene megafauna from the Last
Glacial age or the large land and marine mammals like whales and dolphins. Exhibits on human
evolution and primates are more popular than similarly well-structured exhibits on other species.
We use this to introduce concepts on deep time and the fossil record, with the Chicxulub exhibits
on relations and evolution of the various groups particularly the dinosaurs and mammals.
Dinosaurs and mammals coexisted for a long time, with distinct spatial distributions, habitats, body
masses and lifestyles. What happened after dinosaurs, marine and flying reptiles, ammonites and
many other groups went extinct helps to appreciate macro-evolutionary traits, species
interdependency, how species evolve and interact, how ecosystems develop and function and how
species relate and react to environmental and climatic conditions (Jablonski, 2005, 2008;
Bambach, 2006; Barrett et al., 2009).
The End-Cretaceous mass extinction is the fifth and last mass extinction in the geologic record
(Emiliani et al., 1981; Bambach, 2006). Exhibits on the Phanerozoic extinction events are also
presented, focusing on the marine and land realms, introducing macroevolution and changes
through time (Sepkoski, 1998; Jablonski, 2005, 2008). Adding paleogeographic reconstructions
permits the visualization of the evolving distribution of continents and oceans, particularly the
assemblage of Pangea and its breakup and drift apart, which form the backdrop for life evolution.
How Earth systems interconnect is addressed showing the impact effects on the climate and
environment, with the sharp sudden period of darkness and cooling caused by the fine dust ejecta
in the stratosphere followed by warming due to the massive injection of greenhouse gases (Alvarez
et al., 1980; Alvarez, 1997; Schulte et al., 2010). The deposition of the fine ejecta resulted in
changes in the sea water chemistry, affecting the marine organisms. The warm climates of the
Cretaceous were followed by a cooling trend during the Cenozoic, with the formation of the ice
polar caps and eventually the Late Pleistocene glaciation (Zachos et al., 2008). The evolution of
the different genera, families and species correlates with the changing paleogeography and climate.
Museum visitors often have problems grasping details of evolutionary processes (MacFadden et
al., 2007; Mujtada et al., 2018), which illustrates the challenges particularly for non-formal
curricula and learning outside the classroom. It highlights the role and importance of formal and
informal education and outreach programs, with science museums and supplementary activities
directed to inform and engage on what science is and represents (Stevenson, 1991; Allen, 2004;
Allen and Gutwill, 2004). What is the scientific method and what makes it unique in understanding
the natural world? In recent years with the development of molecular biology, genetics, molecular
clocks and metagenomics, evolutionary studies have entered a new field (Chen et al., 2014).
Introducing new developments and findings presents opportunities and challenges. Recent
discoveries provide unprecedented detail, which allow for a narrative of events, integrating
evidence in a multidisciplinary approach.
**6.3 Geoscience Communication**
Outreach and geosciences communication programs integrate research components with
developments and challenges, reflected in the exhibits, interactive displays and virtual reality
experiences (Louw and Crowley, 2013). Museum exhibits cover a multidisciplinary range of
topics, from the physics of hypervelocity impacts, high pressure/temperature processes and
rheological properties to the delicate balance of geological processes and life evolution. The
museum provides a forum for outreach, educational and science communication, although its
potential needs to be further developed.
The CIRAS addresses matters relevant to policy making and the society. What is critical is having
a better structured relationship with other components of the science park and academic network
and a science communication program with a wide scope and defined priorities (Stewart and Nield,
2013; Stewart and Lewis, 2017). The programs for visiting researchers and postgraduate students,
publications and partnership with the Consortium of Universities for Science expand the academic
program. The CIRAS program includes a weekly seminar series on Chicxulub, mass extinction,
Yucatan and Gulf of Mexico and workshops on technical and science communication themes.
CIRAS conducts geophysical and environmental impact studies, with societal relevance.
Partnership with PCYT research centres and the National Oil Core Repository expands
collaborations and joint activities. Projects in the energy sector that includes oil and gas exploration
in the Gulf of Mexico and on renewable energy are part of the priorities in Yucatan. The joint
projects include laboratory core analyses, geochemistry, petrology, biostratigraphy,
magnetostratigraphy and physical properties, as well as exhibits on oil and gas exploration of the
Gulf and southern Mexico (planned for the Oil Core Repository).
The Chicxulub newsletter, in its fourth year, is published every three months, with notes and
articles on research projects, seminar summaries and news. The Consortium of Universities for
Science formed by institutions in Mexico, the US, the UK and Brazil coordinates the seminar series
with weekly conferences, a science documentary cycle (with discussions by invited specialists),
media interviews and special events. Seminars have addressed Chicxulub drilling projects, life
recovery after the impact, K/Pg mass extinction and after impact radiations. The 2020 seminar
series addressed life evolution, genomics, climate change and health studies, including the Covid-
19 pandemic. Special events include conferences on the Maya civilization, cosmology and
quantum mechanics. The seminars and documentaries are available online in the consortium
platform, which permits a wider use in different countries.
Key aspects for science communication include climate change and effects on biodiversity and
environmental affectation caused by human activity. The global changes affect biodiversity, with
the loss of species that are being interpreted as the sixth mass extinction. Displays showing
examples of how studies connect to life evolution are linked to familiar groups of organisms,
connecting the K/Pg extinction, species evolution and present situation (e.g., Field et al., 2018).
Recent studies on the fossil record and molecular phylogenies are also displayed that show the
intricate interconnections and complex responses during biotic transitions and pre- and post-
extinction processes. A recognized task is effective communication to policy makers and society
(Arattano et al., 2018; Stewart and Lewis, 2017; Illingworth et al., 2018).
7. **Conclusions**
The Chicxulub science museum is built around a unique geological event that marks the transition
of the Mesozoic and Cenozoic Eras. The Earth system science is captured in one place, developing
wide-reaching effective science communication, educational and outreach projects, with a
collaboration network and academic activities. The museum develops from the studies on the
Chicxulub impact, End-Cretaceous mass extinction and Cretaceous/Paleogene boundary and is a
key component of the research complex in the Yucatan Science and Technology Park in Mexico.
The Chicxulub complex is strategic to promote the geosciences in Mexico. It provides the physical
and human capacities, permitting to interconnect research, policy makers and the society. The
museum is an attractive space for learning, exploring and experimenting aimed to engage the
interest of children, youngsters and adults. The research laboratories enhance the capacities,
making it more inviting to learn, wonder and experiment. Science museums are linked to the
development of modern societies, with science and technology being the driving forces for the
transformation of societies.
The complex serves as a hub for multi- and interdisciplinary projects on the Earth and planetary
sciences, climate change and life evolution, fulfilling a recognized task for communication of
geosciences. With the $40^{th}$ anniversary of the impact theory and discovery of the Chicxulub
structure, research on the impact and mass extinction has intensified. In a wide context, enhanced
understanding of the Earth System, processes, life evolution and extinctions and impact of
anthropogenic activities is critical to address the geo-environmental challenges. CIRAS aims to
provide scientific and technical information and advice to society and decision-makers and to
construct a wide collaboration network.
Author Contributions: Authors contributed to the study and in writing the manuscript.
Competing Interests: Authors declare they have no conflict of interest
**Acknowledgments**
We greatly appreciate the constructive revision of the manuscript by Editor Iain Stewart and the
comments by editor Jon Tennant, Christian Koerberl and two anonymous reviewers. CIRAS is a
collaborative effort between the Ministry of Science, Innovation and Higher Education SIIES of
the Yucatan government and the National University of Mexico. We thank the SIIES Secretary
Bernardo Cisneros Buenfil and director Ricardo Bello. The collaboration by the partners in the
project Raúl Godoy Montañez, Fernando D'Acosta, Arcadio Poveda, Enrique Ortiz Lanz, Leon
Faure, Zeus Mendoza, Wilbert Echeverria, Alberto Canto, Inocencio Higuera, Laura Hernández,
Tomas Gonzalez and the Chicxulub group is greatly acknowledged. Raúl Godoy coordinated the
Parque Científico y Tecnológico de Yucatan (Yucatan Science Park, PCYTY). The exhibition in
the Gran Museo de Mundo Maya on the Chicxulub and the Dinosaur Extinction was coordinated
by Enrique Ortiz Lanz.

**List of Figures**

Fig. 1. Chicxulub Center for Scientific Research and Advanced Studies in the Yucatan Science City of the Parque Cientifico y Tecnologico de Yucatan. Views of the Chicxulub research complex, with the museum, laboratories and core repository (photos: J Martinez, Z Mendoza).

Fig. 2. Yucatan Science City (Parque Cientifico y Tecnologico de Yucatan, PCYTY) in Sierra Papacal, Yucatan, Mexico. View to the south of the central PCYTY sector, with the Central Library Building (Drone image, www.pcty.com.mx; Parque Cientifico y Tecnologico de Yucatan).

Fig. 3. Chicxulub crater. (a) Map of Gulf of Mexico and Yucatan peninsula, showing location of the Chicxulub crater. (b) Satellite interferometry radar image of the northern Yucatan peninsula (image courtesy NASA Jet Propulsion Laboratory), showing the surface topographic semi-circular depression above the buried Chicxulub crater rim. Location of the Chicxulub CIRAS center is shown by the star and arrow. Also marked for reference the location of Merida City and Chicxulub Puerto. (c) Chicxulub crater gravity anomaly (Sharpton et al., 1993), showing the concentric semi-circular pattern with the central gravity high and gravity rings marking the peak-ring and multi-ring morphology. (d) Schematic structural model showing the basin, central uplift, terrace zone, melt sheet, breccias and target Cretaceous sediments (Collins et al., 2008).

Fig. 4. (a) Chicxulub drilling programs. View of the drill rig for the Yaxcopoil-1 borehole, core samples for the impact breccias-Paleocene carbonates contact and core repository. (b) View of drilling platform for the Chicxulub IODP-ICDP Expedition 364 drilling project over the peak-ring zone. Marine geophysical surveys, view of the UNAM R/V Justo Sierra. (c) The Cretaceous/Paleogene (K/Pg) boundary is marked globally by the ejecta layer (Schulte et al., 2010). K/Pg boundary sites record a major event in life evolution. In the Gulf of Mexico-Caribbean Sea area the boundary is characterized by high energy sediments in between the basal spherules and clay layers.

Fig. 5. Chicxulub Science Museum in the PCYTY Yucatan Science Park. The Central Library building houses the museum in the second and third floors and views of the exhibits (see also Perez-Cruz and Urrutia-Fucugauchi, 2015).

Fig. 6. Partial view of displays of the exhibition on Chicxulub and the extinction of dinosaurs in the Gran Museo Mundo Maya in Merida, Yucatan.

Fig. 7. Chicxulub laboratories, with view of the six laboratory facilities and some of the instrumental facilities.

Fig. 8. Chicxulub Science Museum and exhibits of the Universe and Solar System.

Fig. 9. Chicxulub Science Museum and exhibits of the Chicxulub crater and impacts.

Fig. 10. Chicxulub Science Museum and exhibits on life evolution and mass extinctions. Exhibits on dinosaurs and other flying and marine reptiles are arranged inside and in the museum surroundings.

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

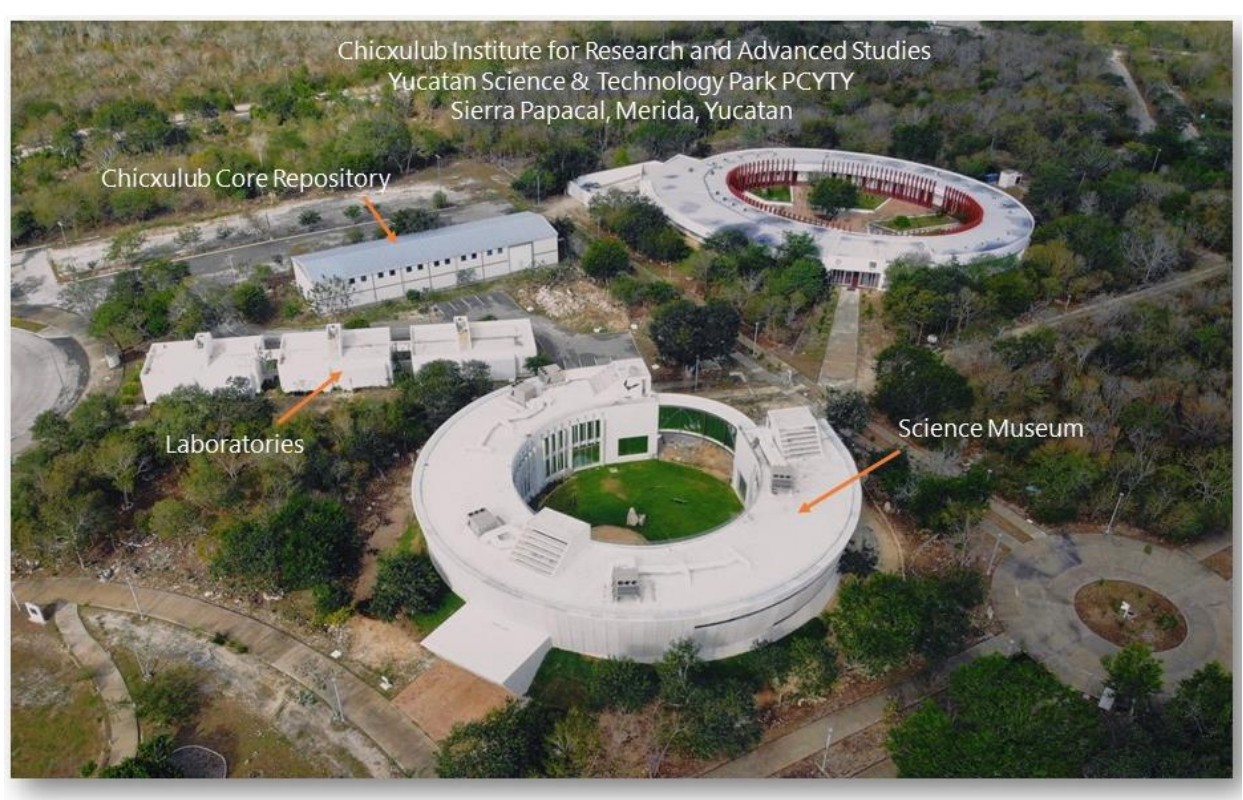

Fig. 1.

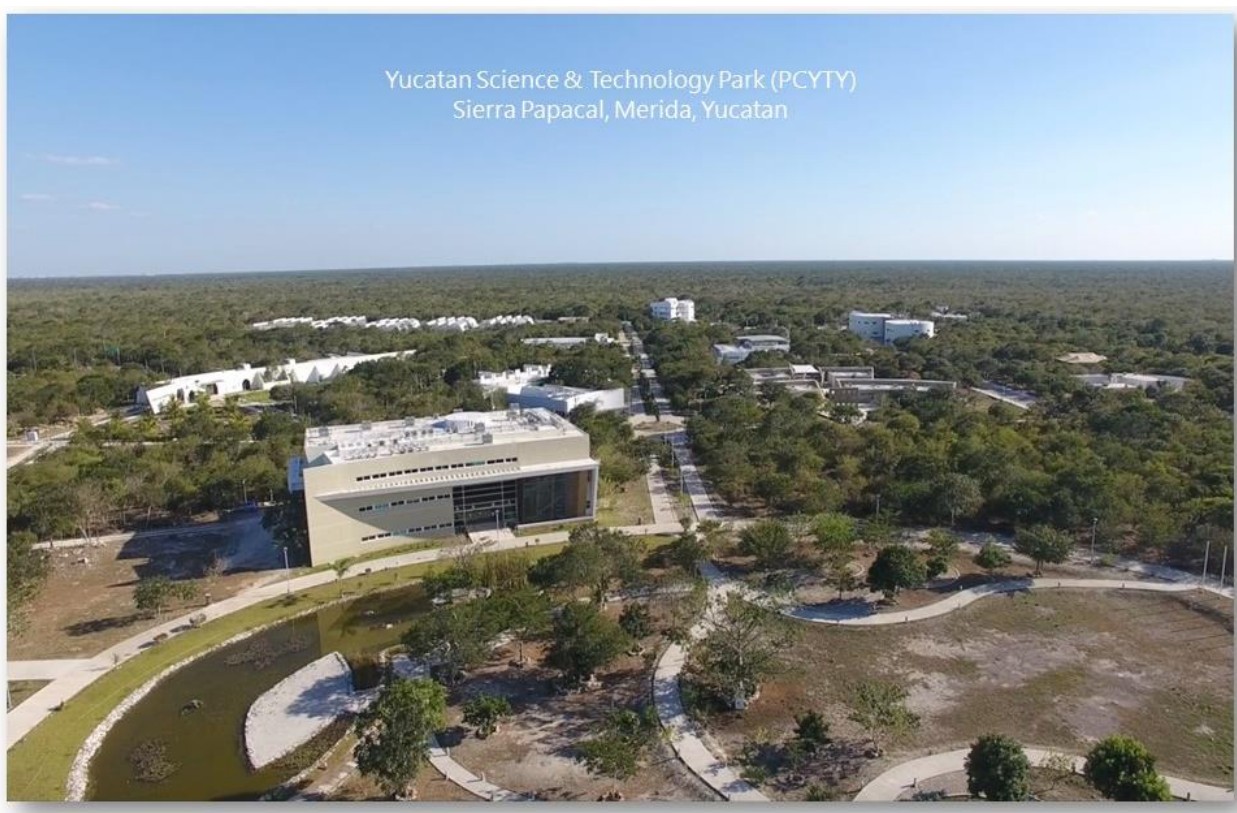

Fig. 2.

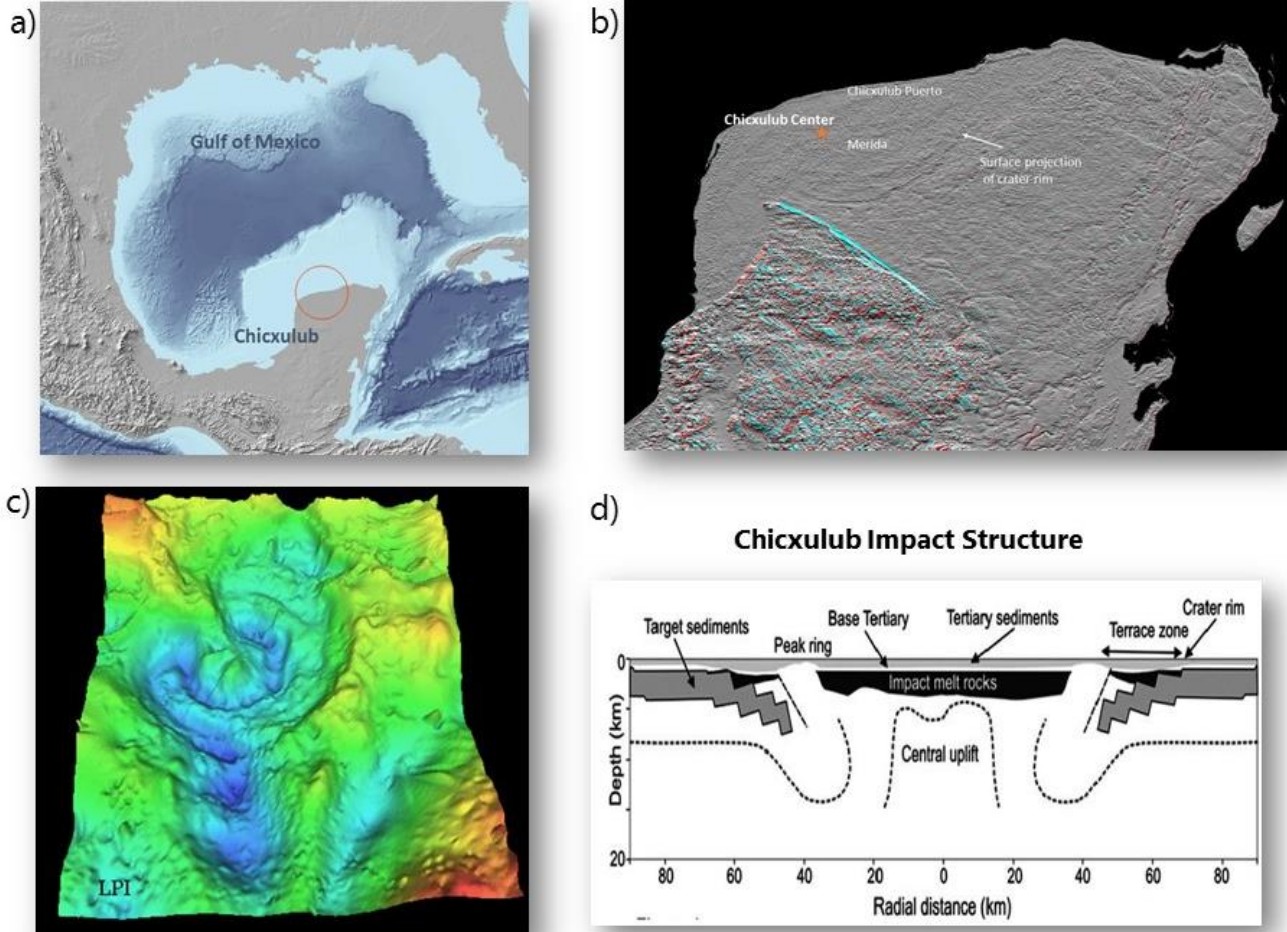

Fig. 3.

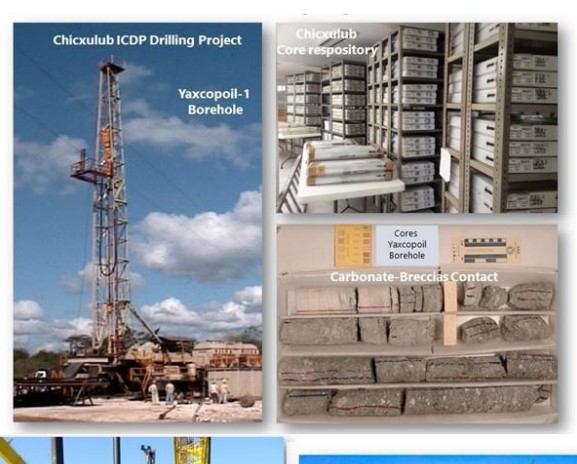

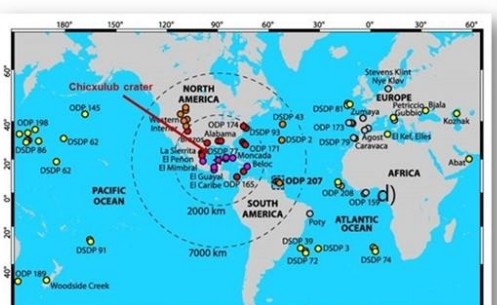

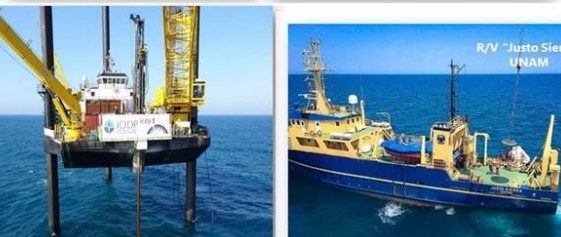

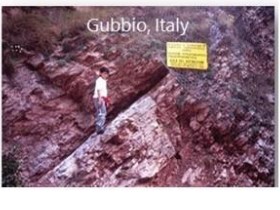

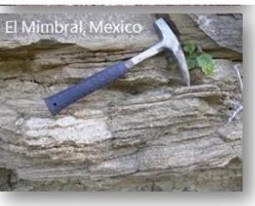


Fig. 4.

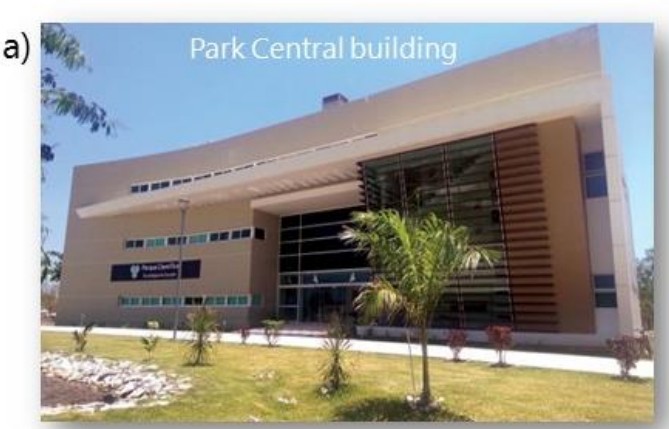

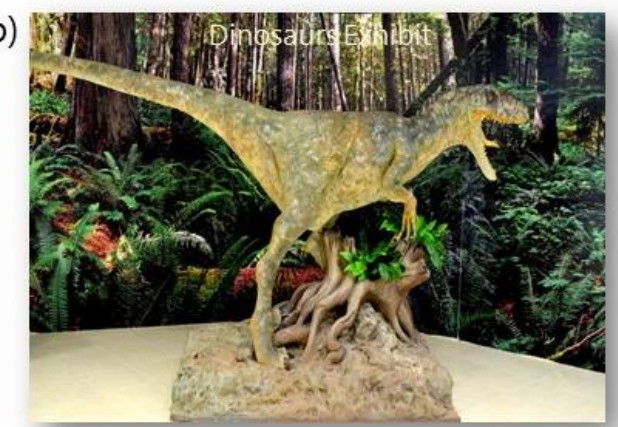

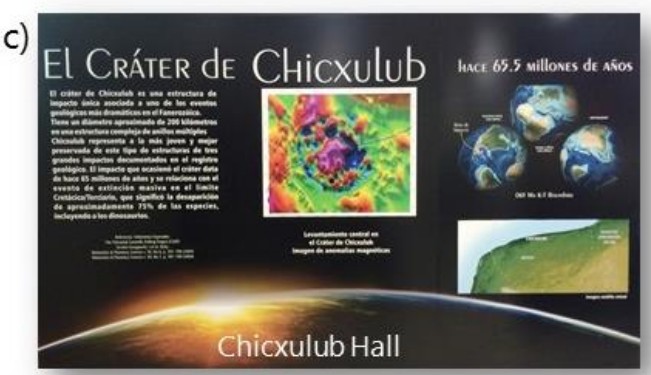

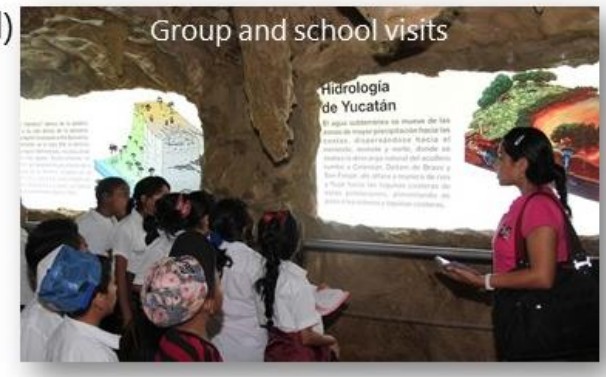


Fig. 5.

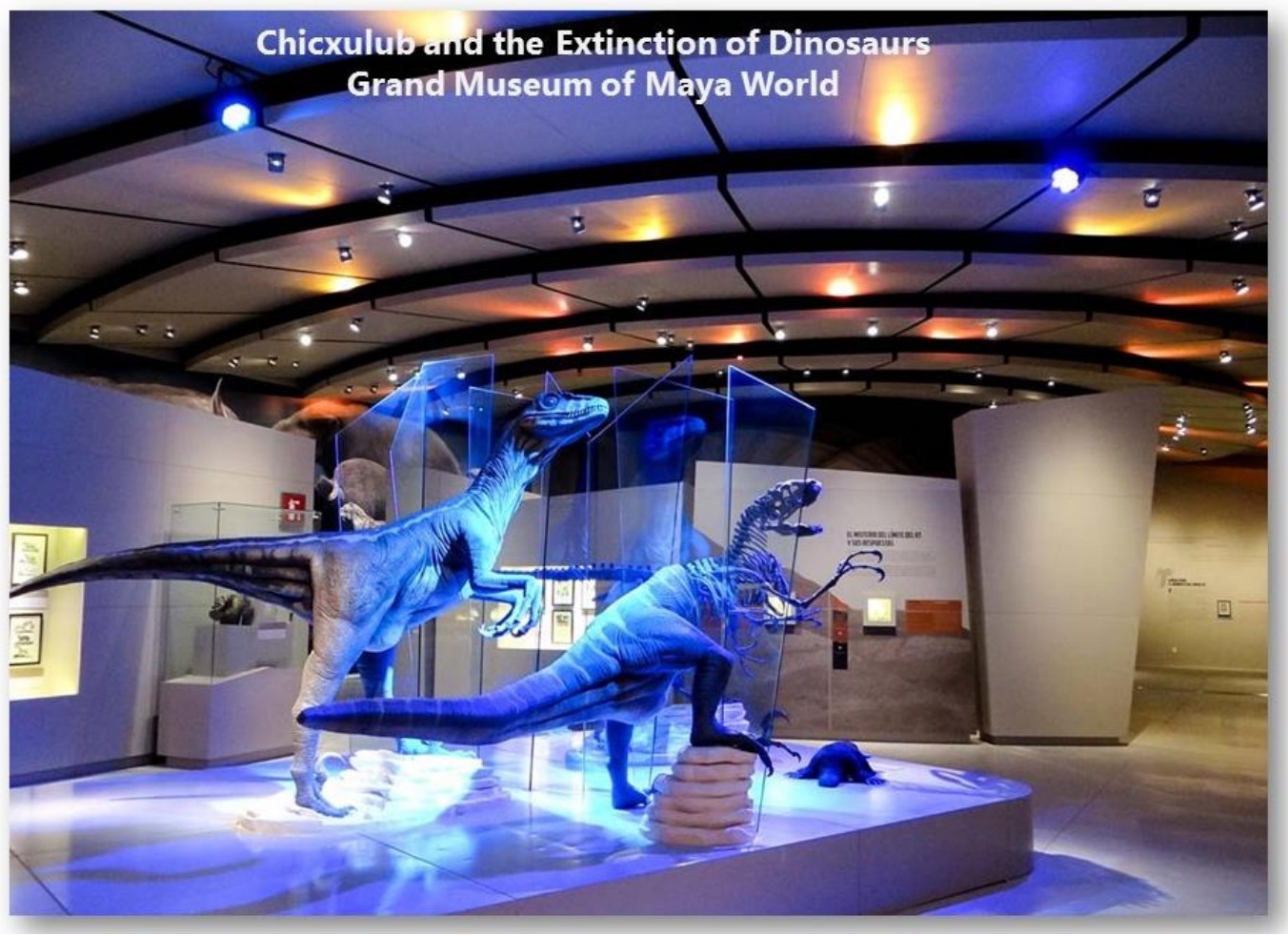


Fig. 6.

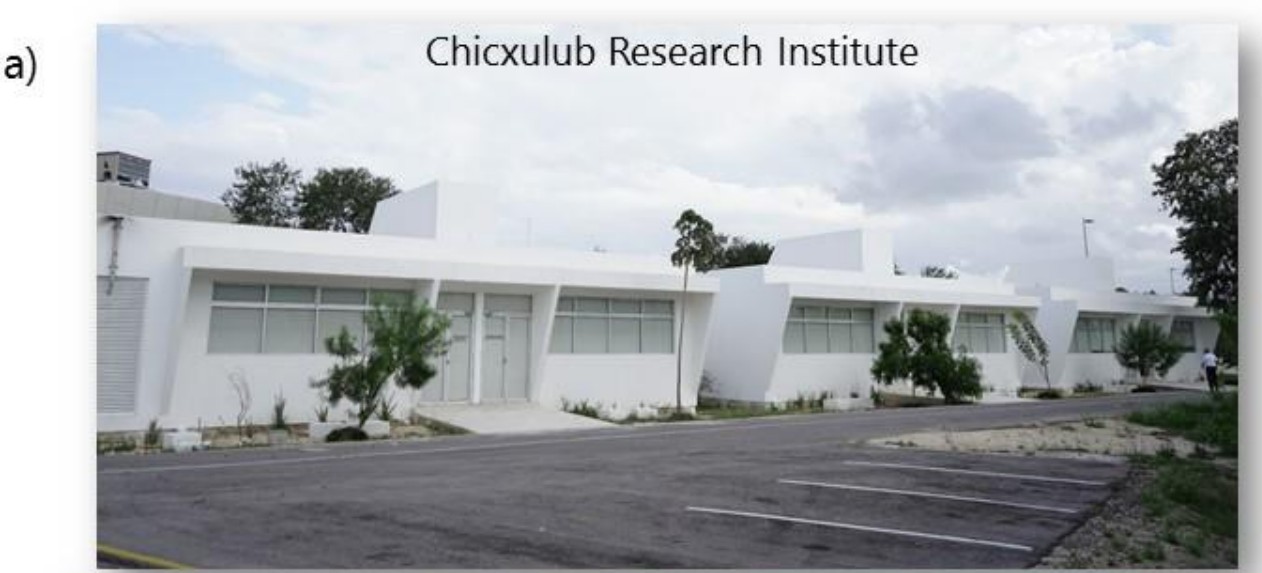

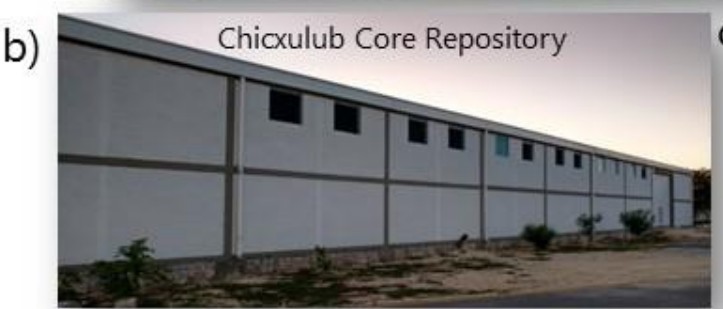

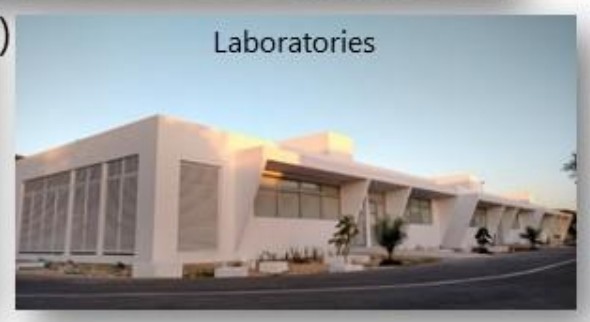

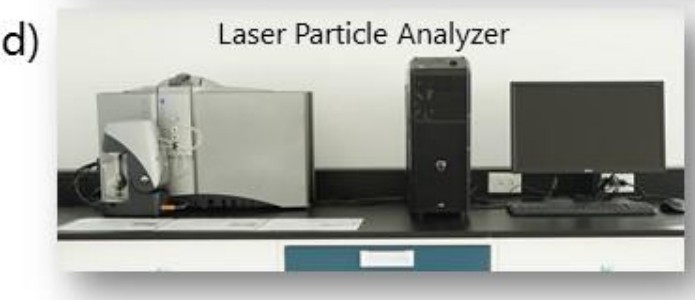

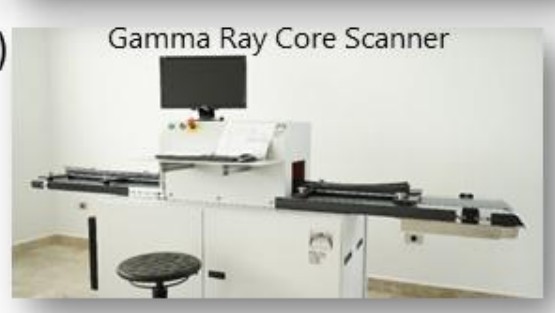

Fig. 7.

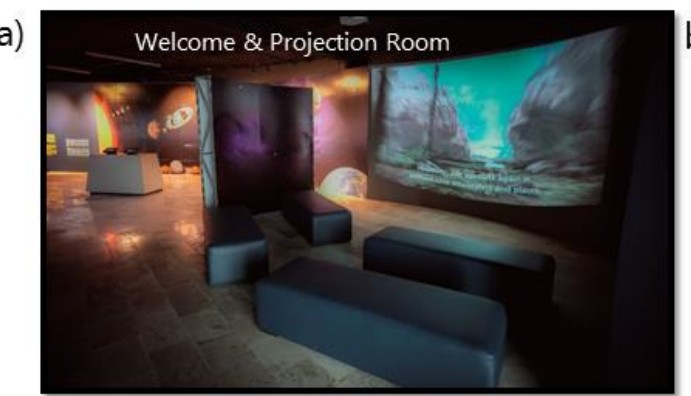

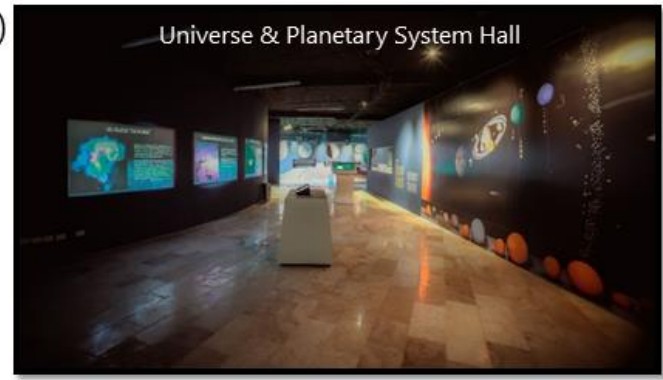

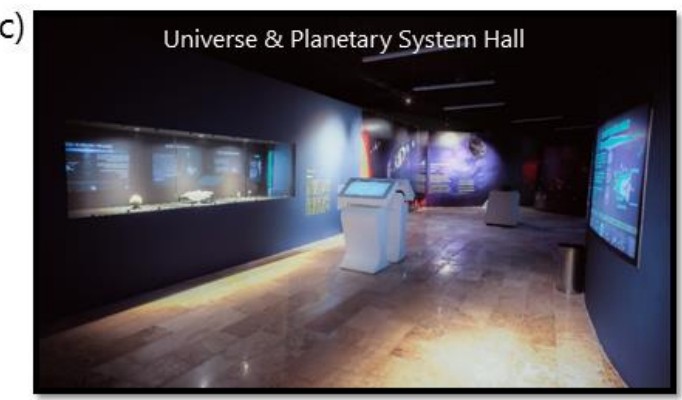

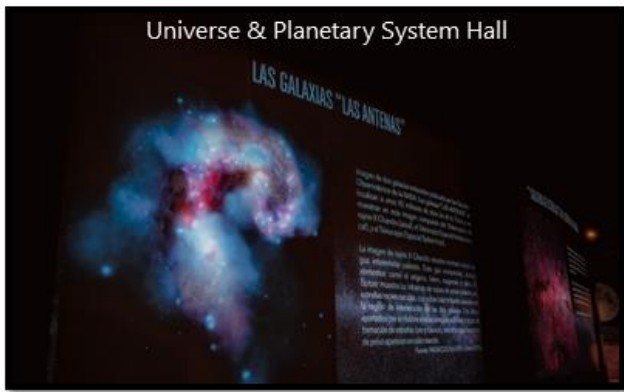

Fig. 8.

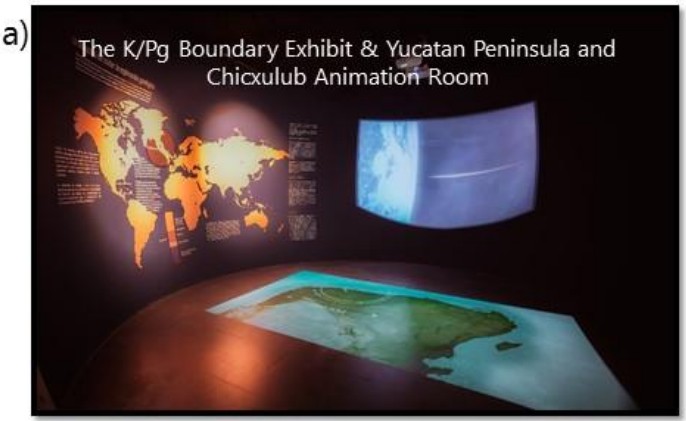

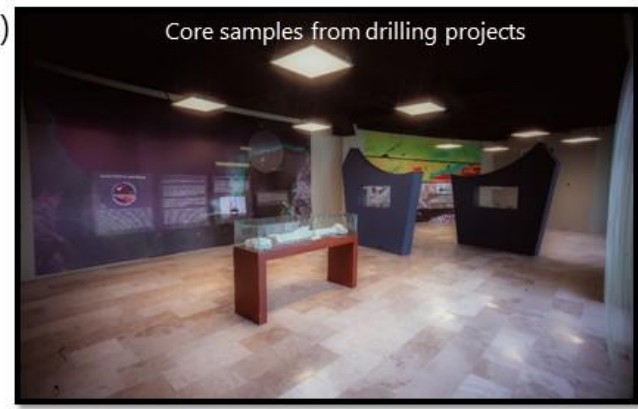

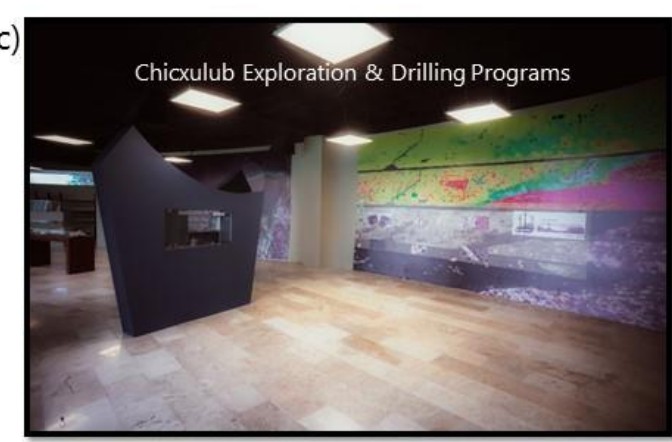

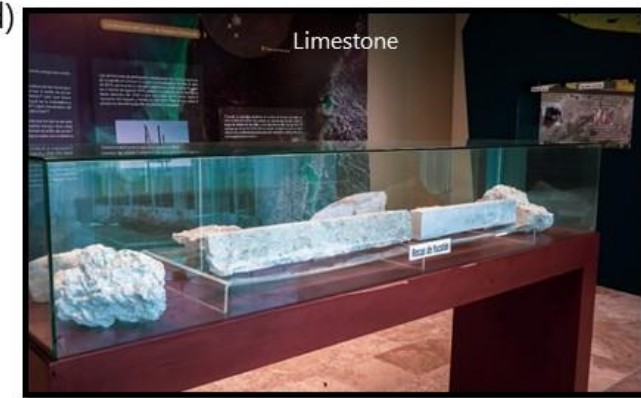

Fig. 9.

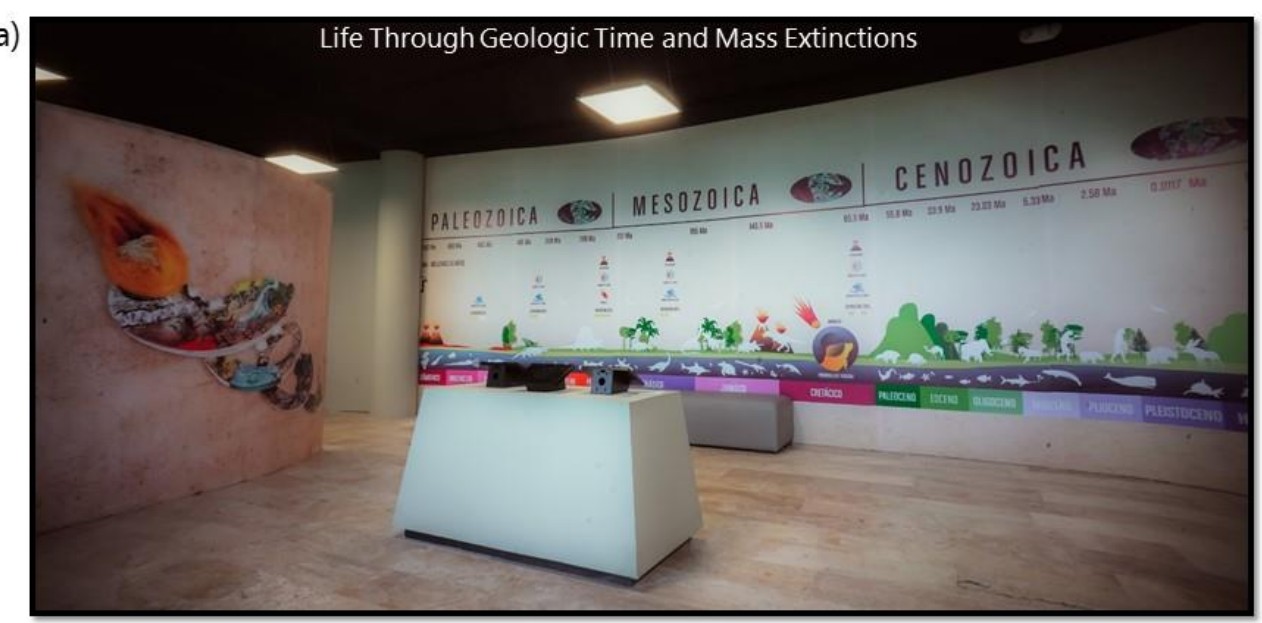

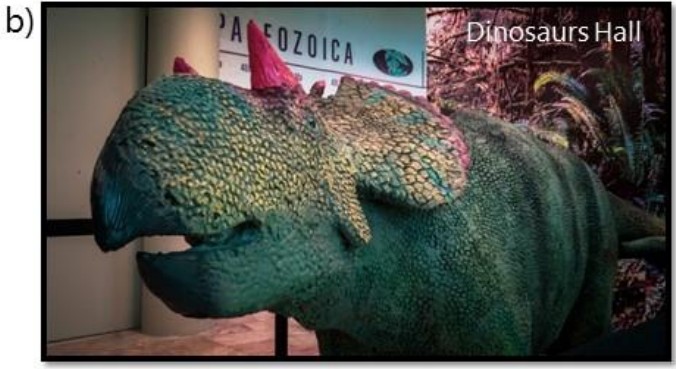

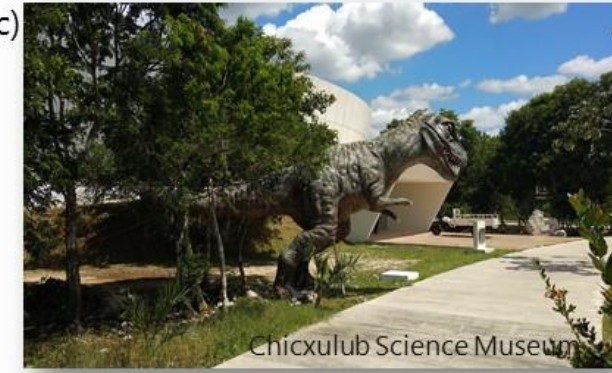

Fig. 10.