# Peer review of "Chicxulub Museum, Geosciences in Mexico, Outreach and Science Communication - Built From the Crater Up"

_Geoscience Communication, 2020_

## Referee Comment (RC1) · Anonymous Referee #1 · 10 May 2020

The manuscript addresses an important topic in science communication – science outreach via geoscience museums. The manuscript describes various activities of the museum and the research center in Yucatan, Mexico built at the place of the Chicxulub asteroid impact. The reading of the manuscript is enjoyable and easy. Meanwhile, the manuscript looks like a written piece of information about the museum and the center, and not as a research article. The paper should be revised to clarify the goal of the study/paper, research questions and methods/approaches, and results obtained.

General Comments

1. Abstract is rather descriptive and not focused on the goal of the study, main research questions, methods, and results of the study. If the research question was to understand how visitors appreciate science related to an asteroid impact and to other

related topics such as extinction and emergence of species, climate change and natural hazards, then it should be mentioned in the abstract and clarified in the manuscript.

2. The paper should be more focused on the research done and results obtained rather than on the description of many details related to the museum and the research center.

3. There are a number of technical issues which should be fixed during the manuscript's revision (see below).

Specific Comments

- Line 18: GeoParks are mentioned in the manuscript in a few places. Do you mean a national geoparks or UNESCO GeoParks? Pls clarify.

- Line 44: Mujtaba et al., 2018 – missing reference

- Line 66: Penfield and Camargo, 1981 – missing reference

- Line 82: Fig. 2 appears at first after Figs. 3-7. The figure numbering should be revised.

- Line 91: "coordinated by Enrique Ortiz Lanz". It should be clarify whether "Enrique Ortiz Lanz" is the name belonging to a person (then professional affiliation) or to company (then professional specialization). Otherwise, it is unclear why Enrique Ortiz Lanz is mentioned here..

- Line 135: it is mentioned that the Chicxulub asteroid impact structure is one of three large impact structures. Pls name two others.

- Lines 182-183: "(Urrutia-Fucugauchi et al., 2004, 2008, 2011) (Figs. 8, 9)". It seems that the reference should be to Figs. 5 and 6.

- Line 235-236: How the topics of climate change, sea level rise and space-related hazards are communicated to the public in the museum? Provide some examples or specific approached of the communication. This may help other museums in the

development of science communication strategies and methods.

- Several publications are included in the Reference List but not referred in the manuscript, namely, Allen (2004), Allen and Gitwill (2004), Dahlstrom (2014), Melosh (1989), Panda and Mohanty (2010), Stevenson (1991), Urrutia-Ficigauchi and Peres-Cruz (2009).

————————————————————

---

## Referee Comment (RC2) · Anonymous Referee #2 · 10 Jul 2020

General comments

The manuscript addresses an interesting and important museum and research centre dedicated to the probably most famous known asteroid impact in the world. It also describes geological and research data, infrastructure aspects and outreach initiatives, as well as educational and tourism use. It also addresses the public support by local government.

However, all this information is randomly distributed along the text, making it difficult to follow the various elements that compound the whole scenario regarding the establishment and the importance of the centre and, moreover, the importance of this kind of museum in the global context.

[Figure]

Mu suggestion is to reorganize completely the manuscript following an order that allows the reader to go through the different aspects starting from basic data regarding the place to the outreach facilities, proposals and comparisons with other places in the world.

Specific comments

Title – The whole paper is based on a specific place – the title should name it.

Introduction – As proposed, the main aim of the paper is to use the example of that specific site museum and research centre to address the role of this kind of place in geoscience outreach. This general contextualization should come first, as well as the information about the relevance of the place.

Items 2, 3 and 4 – Instead of describing these places separately, including facilities, research aspects, information on visitors, and so on, it would be better to describe them according to specific themes. So, the reader would have a complete idea of: 1) How the place is and what kind of facilities it has; 2) What kind of information it shows and its relevance; 3) What kind of public it has; 4) How is the interaction of this public.

From this information, it would be easier to compare it with other exhibitions that are mentioned in the text and finally to discuss their relevance.

No quantitative or, at least, semi-quantitative data are shown regarding the public interaction. It is important to address the discussion.

Discussion - The discussion is confused and the various subjects (relevance of the museum regarding mass extinction and K/Pg boundary, relevance of natural history museums, integration with other aspects such as flora and fauna, common misconceptions, among others) are all mixed with information on specific findings about impact craters and their global importance. I think the prime proposal of the paper (which is in the title) is mixed along the text and did not receive the proper attention.

Conclusions - The conclusions should not be a synthesis of the paper, but contain reflexions and proposals that come from it. Also, normally it does not contain references.

For the figures: some of them are really technical (for example, fig 6 and 10) and should be a minor part of a paper addressing geoscience outreach aspects.

Technical corrections

GeoPark – if this refers to an UNESCO Global Geopark, it should be written with no capital letter

---

## Referee Comment (RC3) · Christian Koeberl (Referee) · 23 Jul 2020

Review of "Built From the Crater Up-Site Museums in Geosciences Communication and Outreach" by UrrutiaFucugauchi et al.

Review by: C. Koeberl, University of Vienna, Austria

The manuscript describes a museum built in Yucatan to give information on the Chicxulub impact structure and event. As such, the text is mostly OK. However, the title of the manuscript promises something different, and more - a discussion of "site museums....". Only one site and only one museum is discussed, namely Chicxulub. Why is there not even any passing mention of, and description of, museums at various other impact sites, such as Meteor Crater (Barringer Crater) in Arizona, USA, the Ries

[Figure]

Crater museum in Noerdlingen, Germany, the Tswaing crater museum in South Africa, the Steinheim crater museum in Germany, and several others? I think to do the topic justice, at least a short section on comparison with other international museums on similar topics should be included.

A few other short comments:

I am missing any information on when the described museum opened, and where to find any visitor information. If it did not yet open (I think the opening was delayed several times, but hopefully it is open by now??) then this should be mentioned, and an opening date should be given, because otherwise, what is the reader to do with information about an inaccessible museum? (The sad story of the museum/visitor center at Vredefort in South Africa comes to mind).

Chicxulub is often referred to as a "crater" but in the international impact community such large and already modified "craters" are usually called an "impact structure".

Some references are in the list but not in the text and/or vice versa. Some more recent publications resulting from the ICDP-IODP drilling should be included.

Regarding the figures, I think to reproduce many old images (such as the gravity map, or photos and logos from the drilling) could be reduced somewhat, and more photos from the actual exhibitions at the museum be included.

Otherwise I think this is a useful contribution and would recommend acceptance after moderate revision.

---

## Author Comment (AC3) · 17 Aug 2020

Thanks very much for the review of the paper. We agree and appreciate the comments by Dr. Christian Koeberl. The title and abstract are modified to make explicit that the paper presents and focuses on the Chicxulub museum. We agree that addressing the museums at various impact sites adds to the presentation and discussion. We thank for calling our attention on this omission. In the revised paper, references to the museums at other impact sites and geological and natural history museums and meteorite collections are now included in the Introduction as well as in the Discussion section.

In the revised manuscript, we have included a section on the museum project back-

ground and development. Dr Koeberl is right that the museum opening has been delayed several times. This has happened for the most recent project phase. The project as described in the paper was originally proposed and developed over a decade and has consisted of several overlapping subprojects that have been completed. The manuscript describes the several phases of exhibitions and development of the Chicxulub project. These include the Chicxulub museum in the Yucatan Science and Technology Park, opened in 2011 by the Yucatan governor, the Exhibition in the Grand Museum of the Maya World on "The Chicxulub Impact and Dinosaur Extinction", inaugurated by the Mexican president and Yucatan governor in December 2012, and the core repository, research laboratories and facilities. The Chicxulub museum in the Park has been open, with programs with schools and the Yucatan Secretary of Education and has attracted considerable numbers of visitors. In the manuscript figures were included showing the exhibitions and visits by school students in the museum; the overlapping project phases were described, and we referred to the laboratories, core repository, research facilities and the Grand Museum of the Maya World exhibition and Chicxulub Museum in the Park. The third phase was completed last year, with the new Museum building construction, and the exhibitions are complete and operational. We realize that this was not described. In the revised manuscript we have now added additional figures showing these museum facilities and exhibitions, linking them to the text, which has been revised to clarify and differentiate how the museum and research facilities have developed over time.

We agree with the comment on the figures, and the addition of figures on the museum exhibition provides a more adequate balance and focus on the museum activities, status and capabilities.

References in the text and reference list have been corrected and references to the recent IODP-ICDP drilling program and recent studies on the mass extinction are included. The recent studies are interesting and relevant to include and have attracted considerable interest in the media.

In addition to the section on background and development of the Chicxulub museum, we have added a short section on the programs and interaction with the Yucatan Secretaries of Education and Science, Innovation and High Education (SIIES), schools and teachers. The Chicxulub Museum and Institute are part of the SIIES. Additional activities include digital and printed materials, drawing contests for school children, conferences and seminars.

The review has been most helpful to revise the mansucript and to comment on the prospects and possibilities.

---

## Author Response (AR1)

Dr. Iain Stewart
Editor
Geoscience Communication
Dear Dr. Stewart,
Enclosed is the revised version of the article on "Built From the Crater Up – Chicxulub Science
Museum, Geosciences Communication and Outreach" submitted for consideration in Geoscience
Communication. We greatly appreciate the comments and suggestions from the journal referees,
which have permitted to improve the presentation and discussion.

For the revision, we have modified the structure of the manuscript, focusing on the research, with
the inclusion and separation of the sections and subsections in the discussion. The title was
modified to better reflect the article contents. The Abstract has been modified and shortened. The
Introduction has been shortened and the aims/goal of the paper included to examine the role of a
museum in outreach, science communication and education. The museum aims to take advantage
of the Chicxulub impact and the End-Cretaceous mass extinction, being located inside the
multiring crater in Yucatan. This allows to introduce a wide range of themes on life evolution,
geological processes, Earth´s systems, extinction and emergence of species, feedback mechanisms,
which are interesting and challenging. The museum forms part of a research complex, which
permits to expand activities to students and visitors, expanding the capabilities and its potential.
The exhibits address present day concerns, for instance, climate change, sea level rise and effects
of anthropogenic activity.

The discussion has been expanded to address the referee´s recommendations focusing on the
museum exhibits and activities. Subsections are added on the education, outreach and science
communication. The figures are improved and figures showing the museum exhibits added,

The specific comments/questions on the manuscript have been corrected and addressed. In the
material for the revision, a pdf file with the modifications highlighted is included.
We thank for the comments and recommendations. We remain,
Sincerely yours,
J Urrutia Fucugauchi
L Perez-Cruz
A O. Urrutia
________________
Anonymous Referee #1
The manuscript addresses an important topic in science communication – science outreach via
geoscience museums. The manuscript describes various activities of the museum and the research
center in Yucatan, Mexico built at the place of the Chicxulub asteroid impact. The reading of the
manuscript is enjoyable and easy. Meanwhile, the manuscript looks like a written piece of
information about the museum and the center, and not as a research article. The paper should be
revised to clarify the goal of the study/paper, research questions and methods/approaches, and
results obtained.

General Comments 1. Abstract is rather descriptive and not focused on the goal of the study, main
research questions, methods, and results of the study. If the research question was to understand
how visitors appreciate science related to an asteroid impact and to other related topics such as
extinction and emergence of species, climate change and natural hazards, then it should be
mentioned in the abstract and clarified in the manuscript. 2. The paper should be more focused on
the research done and results obtained rather than on the description of many details related to the
museum and the research center. 3. There are a number of technical issues which should be fixed
during the manuscript's revision (see below).
Specific Comments - Line 18: GeoParks are mentioned in the manuscript in a few places. Do you
mean a national geoparks or UNESCO GeoParks? Pls clarify. - Line 44: Mujtaba et al., 2018 –
missing reference - Line 66: Penfield and Camargo, 1981 – missing reference - Line 82: Fig. 2
appears at first after Figs. 3-7. The figure numbering should be revised. - Line 91: "coordinated by
Enrique Ortiz Lanz". It should be clarify whether "Enrique Ortiz Lanz" is the name belonging to
a person (then professional affiliation) or to company (then professional specialization).
Otherwise, it is unclear why Enrique Ortiz Lanz is mentioned here.. - Line 135: it is mentioned
that the Chicxulub asteroid impact structure is one of three large impact structures. Pls name two
others. - Lines 182-183: "(Urrutia-Fucugauchi et al., 2004, 2008, 2011) (Figs. 8, 9)". It seems that
the reference should be to Figs. 5 and 6. - Line 235-236: How the topics of climate change, sea
level rise and space-related hazards are communicated to the public in the museum? Provide some
examples or specific approached of the communication. This may help other museums in the
development of science communication strategies and methods.
- Several publications are included in the Reference List but not referred in the manuscript, namely,
Allen (2004), Allen and Gitwill (2004), Dahlstrom (2014), Melosh (1989), Panda and Mohanty
(2010), Stevenson (1991), Urrutia-Ficigauchi and PeresCruz (2009).
Response - Thanks
Thanks very much for the review and comments on the manuscript. We agree with your
recommendations on revising the manuscript to clarify the study goal, research questions,
methods/approaches and results. The research aims of the center and museum are explained in
additional detail. The questions addressed include how to introduce and present the science on the
asteroid impact and End-Cretaceous mass extinction, how to take advantage of this geological site
for attracting and engaging visitors and how visitors, including teachers and students appreciate
these topics and those related to life evolution, geologic processes, climate change and hazards.
The revised manuscript has been reorganized, shortening the descriptive material to focus on the
study and to incorporate the modifications and additions. The Abstract, Discussion and
Conclusions are revised. The conclusions are shortened, focusing on the museum project and
study, and adding comments and proposals.
We also address the specific comments, including the corrections on the Geoparks, figure
numbering and checking the references in the text and reference list. We expand the discussion on
the topics of climate change, sea-level rise and hazards, which are part of the exhibits in the
museum.

Anonymous Referee #2
The manuscript addresses an interesting and important museum and research centre dedicated to
the probably most famous known asteroid impact in the world. It also describes geological and
research data, infrastructure aspects and outreach initiatives, as well as educational and tourism
use. It also addresses the public support by local government. However, all this information is
randomly distributed along the text, making it difficult to follow the various elements that
compound the whole scenario regarding the establishment and the importance of the centre and,
moreover, the importance of this kind of museum in the global context.
My suggestion is to reorganize completely the manuscript following an order that allows the reader
to go through the different aspects starting from basic data regarding the place to the outreach
facilities, proposals and comparisons with other places in the world. Specific comments Title –
The whole paper is based on a specific place – the title should name it. Introduction – As proposed,
the main aim of the paper is to use the example of that specific site museum and research centre to
address the role of this kind of place in geoscience outreach. This general contextualization should
come first, as well as the information about the relevance of the place. Items 2, 3 and 4 – Instead
of describing these places separately, including facilities, research aspects, information on visitors,
and so on, it would be better to describe them according to specific themes. So, the reader would
have a complete idea of: 1) How the place is and what kind of facilities it has; 2) What kind of
information it shows and its relevance; 3) What kind of public it has; 4) How is the interaction of
this public. From this information, it would be easier to compare it with other exhibitions that are
mentioned in the text and finally to discuss their relevance. No quantitative or, at least, semi-
quantitative data are shown regarding the public interaction. It is important to address the
discussion.
Discussion - The discussion is confused and the various subjects (relevance of the museum
regarding mass extinction and K/Pg boundary, relevance of natural history museums, integration
with other aspects such as flora and fauna, common misconceptions, among others) are all mixed
with information on specific findings about impact craters and their global importance. I think the
prime proposal of the paper (which is in the title) is mixed along the text and did not receive the
proper attention.
Conclusions - The conclusions should not be a synthesis of the paper, but contain reflexions and
proposals that come from it. Also, normally it does not contain references. For the figures: some
of them are really technical (for example, fig 6 and 10) and should be a minor part of a paper
addressing geoscience outreach aspects.
Technical corrections GeoPark – if this refers to an UNESCO Global Geopark, it should be written
with no capital letter
Response- Thanks
Thanks very much for the review comments, which are very useful for revising the manuscript.
The comments and recommendations are incorporated in the revision, including the title (". . .-

Chicxulub Museum, Geosciences Communication and Outreach"), mentioning Chicxulub, and the
abstract, introduction and the other sections.
The revision addresses how a Chicxulub science museum can offer interesting opportunities for
presenting and attending outreach, education and geoscience communication. How this unique
geological site can be attractive for engaging visitors and how from this, difficult topics on the
nature of geologic time, life evolution, fossil record, climate change, etc., can be introduced and
how visitors respond to the exhibits and related activities.
The specific comments are taken into consideration for revising the manuscript, reordering the
way of presenting the museum and research facilities. We add a section providing the background
and development of the project and then sections on the presentation on the facilities, exhibits,
interactive activities and how visitors, including teachers, students are considered.
The reorganization allows to present a comparison with other natural history and geological
museums and to discuss advantages and relevance of this museum that focuses on the last major
mass extinction and the Cretaceous/Paleogene boundary. Based on your suggestions, the
discussion has been restructured into three separate subsections to address the different topics, in
this way, the text gives an orderly and easier reading. Thanks. This also facilitates to discuss how
interesting yet difficult concepts are presented, how visitors respond and what strategies and
alternatives could be considered. The information on visitors is semi-quantitative. The exhibition
on the Chicxulub impact and extinction of dinosaurs" in the Grand Museum of Maya World
attracted a larger number of visitors. The "Chicxulub Museum" in the Yucatan Science and
Technology Park attended school groups, teachers and researchers, as well as visitors.
Results from related activities are also addressed in the revision. This includes information/results
of the museum printed material/publications, interaction with teachers and schools, including two
GIFT (Geosciences Information for Teachers) Workshops of the European Geosciences Union
held in Merida in 2010 and 2016. The Panamerican GIFT Workshop, part of the new capacity-
building program of EGU is scheduled for October 2020 to be held in the Chicxulub Museum in
the Yucatan Science and Technology Park. Plans are affected by the worldwide pandemic of
coronavirus (COVID-19), but the program is being reprogrammed. Other interesting activities
included conferences, seminars and drawing contests for school children in primary schools. The
GIFT Workshops have been organized in collaboration with the Secretaries of Education and
Science, Innovation and High Education (SIIES) of the Yucatan government, Universities,
Mexican Academy of Sciences and scientific societies. The Chicxulub Institute and Museum are
part of the SIIES, which permits close interaction with the research and educational system.
The conclusion section has been shortened with conclusions rewritten and proposals to expand and
optimize outreach, educational and science communication activities added. The Chicxulub impact
and extinction of the dinosaurs and other species generate considerable interest on their own and
being the museum in Yucatan - the impact site – opens interesting opportunities for outreach and
geoscience communication.
Thanks for the comment on the figures. We agree and new figures are added on the museum
exhibitions and facilities.

Referee #3
Review of "Built From the Crater Up-Site Museums in Geosciences Communication and
Outreach" by Urrutia Fucugauchi et al.
Review by: C. Koeberl, University of Vienna, Austria
The manuscript describes a museum built in Yucatan to give information on the Chicxulub impact
structure and event. As such, the text is mostly OK. However, the title of the manuscript promises
something different, and more - a discussion of "site museums....". Only one site and only one
museum is discussed, namely Chicxulub. Why is there not even any passing mention of, and
description of, museums at various other impact sites, such as Meteor Crater (Barringer Crater) in
Arizona, USA, the Ries Crater museum in Noerdlingen, Germany, the Tswaing crater museum in
South Africa, the Steinheim crater museum in Germany, and several others? I think to do the topic
justice, at least a short section on comparison with other international museums on similar topics
should be included.
A few other short comments: I am missing any information on when the described museum
opened, and where to find any visitor information. If it did not yet open (I think the opening was
delayed several times, but hopefully it is open by now??) then this should be mentioned, and an
opening date should be given, because otherwise, what is the reader to do with information about
an inaccessible museum? (The sad story of the museum/visitor center at Vredefort in South Africa
comes to mind). Chicxulub is often referred to as a "crater" but in the international impact
community such large and already modified "craters" are usually called an "impact structure".
Some references are in the list but not in the text and/or vice versa. Some more recent publications
resulting from the ICDP-IODP drilling should be included.
Regarding the figures, I think to reproduce many old images (such as the gravity map, or photos
and logos from the drilling) could be reduced somewhat, and more photos from the actual
exhibitions at the museum be included.
Otherwise I think this is a useful contribution and would recommend acceptance after moderate
revision.
Response - Thanks
Thanks very much for the review comments, which are very useful for revising the manuscript.
The comments and recommendations are incorporated in the revision, including the title (". . .-
Chicxulub Museum, Geosciences Communication and Outreach"), mentioning Chicxulub, and the
abstract, introduction and the other sections.
The revision addresses how a Chicxulub science museum offers interesting opportunities for
presenting and attending outreach, education and geoscience communication. How this unique
geological site can be attractive for engaging visitors and how from this, difficult topics on the
nature of geologic time, life evolution, fossil record, climate change, etc., can be introduced and
how visitors respond to the exhibits and related activities.

The specific comments are taken into consideration for revising the manuscript, reordering the
way of presenting the museum and research facilities. We add a section providing the background
and development of the project and then sections on the presentation on the facilities, exhibits,
interactive activities and how visitors, including teachers, students are considered. The
reorganization allows to present a comparison with other natural history and geological museums
and to discuss advantages and relevance of this museum that focuses on the last major mass
extinction and the Cretaceous/Paleogene boundary. Based on your suggestions, the discussion has
been restructured into three separate subsections to address the different topics, in this way, the
text gives an orderly and easier reading. Thanks. This also facilitates to discuss how interesting
yet difficult concepts are presented, how visitors respond and what strategies and alternatives could
be considered. The information on visitors is semi-quantitative. The exhibition on the Chicxulub
impact and extinction of dinosaurs" in the Grand Museum of Maya World attracted a larger
number of visitors. The "Chicxulub Museum" in the Yucatan Science and Technology Park
attended school groups, teachers and researchers, as well as visitors.
Results from related activities are also addressed in the revision. This includes information/results
of the museum printed material/publications, interaction with teachers and schools, including two
GIFT (Geosciences Information for Teachers) Workshops of the European Geosciences Union
held in Merida in 2010 and 2016. The Panamerican GIFT Workshop, part of the new capacity-
building program of EGU is scheduled for October 2020 to be held in the Chicxulub Museum in
the Yucatan Science and Technology Park. Plans are affected by the worldwide pandemic of
coronavirus (COVID-19), but the program is being reprogrammed. Other interesting activities
included conferences, seminars and drawing contests for school children in primary schools. The
GIFT Workshops have been organized in collaboration with the Secretaries of Education and
Science, Innovation and High Education (SIIES) of the Yucatan government, Universities,
Mexican Academy of Sciences and scientific societies.
The Chicxulub Institute and Museum are part of the SIIES, which permits close interaction with
the research and educational system. The conclusion section has been shortened with conclusions
rewritten and proposals to expand and optimize outreach, educational and science communication
activities added. The Chicxulub impact and extinction of the dinosaurs and other species generate
considerable interest on their own and being the museum in Yucatan - the impact site – opens
interesting opportunities for outreach and geoscience communication.
Thanks for the comment on the figures. We agree and new figures are added on the museum
exhibitions and facilities.

[revised manuscript text omitted]

Fig. 1

[Figure]

[Figure]

Fig. 2

[Figure]

[Figure]

Fig. 3

[Figure]

[Figure]

Fig. 4

[Figure]

Chicxulub drilling programs

Chicxulub Marine Geophysics and Drilling Programs

[Figure]

IODP ICDP Chicxulub Drilling Expedition 364

R/V "Justo Sierra" UNAM

Fig. 5

[Figure]

[Figure]

Fig. 6

[Figure]

Gubbio, Italy

El Mimbral, Mexico

[Figure]

Fig. 7

[Figure]

[Figure]

Chicxulub Science Museum

[Figure]

[Figure]

Fig. 8

[Figure]

Chicxulub and the Extinction of Dinosaurs
Gran Museo de Mundo Maya

[Figure]

Fig. 9

[Figure]

Chicxulub Laboratories

[Figure]

Gamma Ray Core Scanner

[Figure]

Laser Particle Analyzer

[Figure]

[Figure]

[Figure]

[Figure]

Fig. 10

[Figure]

[Figure]

[Figure]

[Figure]

Fig. 11

[Figure]

[Figure]

[Figure]

Fig. 12

---

## Author Response (AR2)

**Dr. Iain Stewart**
**Geoscience Communication**
**Executive Editor**

Dear Dr. Stewart,

Thanks very much for the comments on the article, which are most useful in preparing the revised version. Following the recommendations, title is modified to better reflect the article contents and we have rewritten the abstract and conclusions. The questions addressed in the article are better presented and developed in the discussion and conclusions. The modifications include revising the text, adding and comments on the broad context, addition of a section on geosciences in Mexico, and revision of the article structure/organization.

The museum and outreach, educational and science communication activities are better linked and presented, highlighting their relation to the Chicxulub impact, mass extinction and K/Pg boundary. The Chicxulub complex is in this context unique, devoted to an event representing a turning point in the planet`s history with the transition of the Mesozoic and Cenozoic eras. We now rephrased this aspect that has been the overarching theme in the design and planning of the Chicxulub museum, which is better highlighted. The text and figures are revised, adding some parts and condensing and editing others. Figures presenting the museum, aspects of the research complex and laboratories are added, including those on the museum exhibits; we agree that they were need in the manuscript. The revised manuscript takes into consideration the reviewer`s comments and suggestions, including the specific comments. These and your editorial comments and recommendations have permitted to improve the manuscript and correct the organization and presentation.

The review comments have been most helpful, we greatly appreciate the reviews and recommendations. Enclosed is the revised manuscript.

Sincerely yours
Jaime, Ligia, Araxi

**Editor Decision: Publish subject to minor revisions (further review by editor)** (23 Nov 2020) by Iain Stewart

Comments to the Author:

Firstly, apologies for the delay in getting back to you regarding your responses to the three referee reports. I hope it has not caused you inconvenience, though I imagine you are keen to prepare your revised manuscript.

In that regard, I note the generally positive and encouraging comments of all three reviewers, and the carefully considered and specific suggestions that they have to improve the paper. I am gratified that most of these healthy criticisms seem to have been acknowledged in your responses to the reviewers, and it seesm clear that your manuscript will be revised around

them. Some of the reviewers have suggested alternative ways to structure the paper and while these seem sensible it is up to you as to what degree you adopt those suggestions. However, I would ask that the revised paper should aim to address the following overarching points:

(1) A more focused title and abstract that better represents the revised content

Title has been modified

(2) A clearer research question, or at least a research spine to the paper, that more clearly addresses the science of asteroid impact.

Research questions added. The relation of the Chicxulub impact and mass extinction to the science museum and what makes the museum unique and connections to Earth system components are addressed.

(3) A broader context that briefly sets the scene of the museum into the wider perspective of other impact crater museums or visitor centres.

The broad context is added and elaborated in more detail, providing the context for the museum and research complex

(4) A more judicious use of figures and images to reduce the number of old and already published mages and present new images of the actual exhibitions.

This is added, with figures on the museum and research complex and figures on the exhibits.

I am sure that, from your comments, many of these changes are already in preparation, and on that basis I very much look forward to receiving the revised paper.

Kind regards,

Iain

---

## Author Response (AR3)

**Dr. Sam Illingworth**
**Geoscience Communication**
**Chief Executive Editor**

Dear Dr. Illingworth,

Thanks very much for your message and editorial decision on the article. Thanks for the detailed review process; it has been a very constructive experience. We greatly acknowledge the reviews and comments.
Enclosed is the revised version of the article.
Sincerely yours,
Jaime Urrutia-Fucugauchi
Ligia Perez-Cruz
Araxi O. Urrutia

**Executive Editor Decision: Publish subject to technical corrections** (26 Mar 2021) by Sam Illingworth
Comments to the Author: Thank you for choosing Geoscience Communication for your research and for engaging so thoroughly in the peer review process. Once you have made the technical corrections that have been outlined by the Editor, we will be delighted to accept your manuscript for publication.

Many Thanks,
Sam Illingworth
(Chief Executive Editor of Geoscience Communication)

**Dr. Iain Stewart**
**Geoscience Communication**
**Executive Editor**

Dear Dr. Stewart,
Thanks very much for your detailed constructive reviews and comments on the article, which are most helpful. Enclosed is the revised manuscript, incorporating the corrections and changes.
We also greatly appreciate the journal reviewers for their revisions and recommendations.
Sincerely yours
Jaime, Ligia, Araxi

**Editor Decision: Publish subject to technical corrections** (26 Mar 2021) by Iain Stewart
Comments to the Author:
line 22 'allow the integration of up-to-date'
line 40 - 'requires the development and implementation of'
line 45 - 'have a rich tradition in which collections of … play an important role…'
line 49 'allow the integration of'
line 52 '…of university education programmes and of profssional workshops…'
line 53 - 'Global Geoparks'

line 57 - 'Some, such as...'
line 69 - 'Here, we outline how it has develop and examine the potential .... offer, , including the challenges ahead'.
line 77 - 'popular themes like meteorite impacts... atrractive contexts for geoscience engagement'
line 79 - 'have developed public outreach projects'
line 94 - 'the physical and human capabilities that allow the expansion of ...'
line 107 - '...might be expected to be...'
line 122 - 'The Chicxulub...'
line 136 - ADD a closing sentence, such as 'So, in summary, Chicxulub presents an opportunity to showcase the holistic and integrated nature of Earth system science.'
line 143 - 'the plan.... its effects....'
line 182 - UNAM - state in full.
line 228 'that permit issues such as present-day global warming.... to be addressed.'
line 290 - 'In Mexico, research projects address societal issues such as ... but geoscientists have yet to have long-term programs that have an effective public, educational, and policy impact. The CIRAS offers a potenial facility for doing that.'
line 331 - ' ...outcrops further expands...'
line 333 'smartphone applications' ?? [rather than 'apps]
line 378 - '...permits the visualization of...'
line 395 - 'evolutionary studies have entered...'
line 407 - 'What is critical is having a closer...'
line 433 - MAKE CLEAR WHERE THIS QUOTE COMES FROM - WHICH REF??